# Methodological Considerations for Studies Evaluating Bleeding Prediction Using Hemostatic Point-of-Care Tests in Cardiac Surgery

**DOI:** 10.3390/jcm13226737

**Published:** 2024-11-08

**Authors:** Mirna Petricevic, Klaus Goerlinger, Milan Milojevic, Mate Petricevic

**Affiliations:** 1University Department of Health Studies, University of Split, 21000 Split, Croatia; petricevic.mirna@gmail.com; 2Department of Anesthesiology and Intensive Care Medicine, University Hospital Essen, University Duisburg-Essen, 45127 Essen, Germany; kgoerlinger@werfen.com; 3Medical Department, Tem Innovations, 80331 Munich, Germany; 4Department of Cardiac Surgery and Cardiovascular Research, Dedinje Cardiovascular Institute, 101801 Belgrade, Serbia; mln.milojevic@gmail.com; 5School of Medicine, University Hospital of Split, University of Split, 21000 Split, Croatia

**Keywords:** bleeding, chest tube output, transfusion, cardiac surgery

## Abstract

A certain proportion of patients undergoing cardiac surgery may experience bleeding complications that worsen outcomes. Numerous studies have investigated bleeding in cardiac surgery and some evaluate the role of hemostatic point-of-care tests in cardiac surgery patients. The prevalence of excessive bleeding varies in the literature, and such variability stems from the lack of a standardized definition of excessive bleeding. Herein, we report numerous definitions of excessive bleeding and methodological considerations for studies evaluating bleeding using hemostatic point-of-care tests in cardiac surgery patients. We evaluated the role of hemostatic point-of-care devices in contemporary research on bleeding complications and hemostatic management in cardiac surgery. The type of studies (prospective vs. retrospective, interventional vs. observational), patient selection (less complex vs. complex cases), as well as data analysis with comprehensive statistical considerations have also been provided. This article provides a comprehensive insight into the research field of bleeding complications in cardiac surgery and may help readers to better understand methodological flaws and how they influence current evidence.

## 1. Introduction

### 1.1. The Relevance of Bleeding Complications in Cardiac Surgery

A negligible proportion of patients undergoing cardiac surgery may experience significant chest tube drainage (CTD) requiring intervention ranging from transfusion of blood components to re-exploration for excessive bleeding [1,2,3,4]. Dixon et al. [5] showed that CTD was an independent predictor of mortality in patients undergoing cardiac surgery [5]. Furthermore, an independent association between excessive bleeding and adverse clinical outcomes has been reported by Chrisensen et al. [6]. Also, Rannuci et al. described a significant association between postoperative bleeding and operative mortality [7]. This association was further enhanced by the administration of red blood cells (RBC) and to a lesser extent with preoperative anemia [7]. Furthermore, Petrou et al. reported a dose-dependent effect of RBC, plasma, and platelet transfusion on mortality in cardiac surgery [8].

Excessive postoperative bleeding is a relatively common complication in cardiac surgery occurring in about 20% of patients [9] and remains to be a persistent complication of cardiac surgery procedures over a long period.

Re-exploration for excessive bleeding occurs in 5–9% of all unselected cardiac surgical patients [3] and remains a very important complication associated with patient mortality rates up to 14% [3,10,11]. It seems that the risk of an adverse outcome parallels the time for re-exploration [12]. Furthermore, the multicausality of excessive postoperative bleeding makes it somehow challenging to precisely understand the underlying mechanisms. The mechanisms of hemostatic disorders in patients undergoing cardiac surgery with cardiopulmonary bypass (CPB) have recently been described [11]. Hemostatic disorder includes different mechanisms such as hyperfibrinolysis, heparin anticoagulation, protamine overdose, dilution and consumption of fibrinogen, and other coagulation factors, as well as thrombocytopenia and platelet dysfunction [11,13,14,15,16]. Numerous clinical factors contributing to post-CPB bleeding are well recognized [1]; however, not all patients with risk factors will bleed (low positive predictive value), nor all patients without known clinical risk factors will not bleed.

Additional preoperative factors should inevitably be considered, such as platelet dysfunction due to preoperatively administered antiplatelet therapy (APT) or impaired thrombin generation due to oral anticoagulation with vitamin K-antagonists (VKAs) or direct oral anticoagulants (DOACs). Furthermore, an acquired von Willebrand syndrome must be considered in patients with structural heart disease (e.g., aortic stenosis) resulting in high shear stress, which increases ADAMTS13 mediated cleavage of von Willebrand factor multimers. All these factors, either alone or in combination, cause hemostatic disbalance, which in turn may contribute to bleeding complications.

### 1.2. The Relevance of Transfusion Requirements in Cardiac Surgery

Adult cardiac surgery accounts for up to 20% of all RBC transfusion in the Unites States [17,18,19,20,21].

It is obvious that cardiac surgery patients receiving blood transfusion are at risk for increased mortality and morbidity, and that the transfusion practice expresses huge variability with a widespread range of transfusion triggers. Shaw et al. attempted to elucidate whether this increased risk occurs across all hematocrite (HCT) levels [22]. The authors concluded that patients who received blood at all HCT levels may be placed at an increased risk of operative mortality and/or other surgical complications [22]. Unexpectedly, transfusion was still associated with a higher complication and mortality rate even in patients with low HCT [22]. A lack of benefit of transfusing patients at all HCT levels was noted [22].

In a study by Ranucci et al. [7], they showed the deleterious effect of major bleeding had a multiplying effect if coexisting with RBC transfusion and to a lesser extent, preoperative anemia [7]. The importance of the three aforementioned factors may be easily understood by the information that patients without preoperative anemia or major bleeding and not being transfused have a very low operative mortality rate from 0.6% to 0.7% [7]. This inevitably leads to the obvious consideration that controlling these risk factors may be of paramount importance in cardiac surgery.

Considering the impact of bleeding and transfusion on clinical outcomes, determining which of these factors is the predominant driver of adverse outcomes is paramount to advancing our knowledge regarding perioperative bleeding management.

### 1.3. The Role of Hemostatic Point-of-Care Devices in Contemporary Research of Bleeding Complications and Hemostatic Management in Cardiac Surgery

Despite the increased understanding of the hemostatic disorder mechanism, the question “How to predict and prevent excessive bleeding?” remains challenging. Conventional coagulation tests failed to predict postoperative bleeding [23]. Furthermore, long turnaround times from blood sampling to obtaining results make conventional laboratory tests to be of limited value in the clinical decision-making process. The complexity of bleeding mechanisms hampers the possibility of simple detection and treatment, and rather requires a comprehensive approach.

It sounds reasonable that hemostatic point of care testing (HPOCT) may help to better understand the underlying mechanisms of hemostatic disorders, and therefore, improve diagnostic performance and accuracy in predicting bleeding complications in cardiac surgery.

Since platelets may play a pivotal role in hemostatic disorders in cardiac surgery and the importance of short turnaround times, several HPOCT devices have been developed.

These point-of-care hemostatic tests may be divided into the following: (1) Viscoelastic tests such as ROTEM (TEM Innovations, Munich, Germany) and TEG (Haemonetics Corporation, Braintree, MA, USA); and (2) Platelet function analyzers based on either whole blood impedance aggregometry [Multiplate—(Roche Diagnostics, Mannheim, Germany), ROTEM platelet—(TEM Innovations, Munich, Germany)], whole blood light transmission (VerifyNow, Werfen, San Diego, CA, USA), viscoelastic testing (TEG Platelet Mapping, Haemonetics Corporation, Braintree, Massachusetts, United States), or shear stress aggregometry (PFA—200, Siemens GmbH, Munich, Germany).

Both platelet function testing and viscoelastic testing, have been used in numerous trials to assess the predictability of bleeding complications; however, contradictory data exists which hampers the pooling of the evidence and drawing clear conclusions on the role of these devices in everyday clinical hemostatic management. The heterogeneity of reported data is further corroborated by different study settings with different non-standardized endpoints assessed, along with different devices used.

### 1.4. Contemporary Management of Hemostasis in Cardiac Surgery: What Do Guidelines Say?

The American Society of Anesthesiologists Task Force of Perioperative Blood Management came out with updated practice guidelines for perioperative blood management [24]. These guidelines include a greater emphasis on the preoperative assessment of patients and adjuvant therapies such as drugs and techniques to reduce or prevent bleeding complications and transfusion requirements [24]. For intraoperative and postoperative monitoring of hemostasis, these guidelines recommend that point-of-care platelet function testing and/or viscoelastic testing (e.g., TEG or ROTEM) may be included in the monitoring of perioperative hemostasis [24].

The European Society of Anesthesiology published guidelines on the management of severe perioperative bleeding [25]. These guidelines recommend the use of transfusion algorithms based on HPOCT for the evaluation of coagulation status [26]. Evaluation of platelet function is suggested for patients with positive bleeding history, medical conditions that may result in platelet dysfunction, or patients exposed to APT prior to surgery [26]. The guidelines [26] do not support the routine use of platelet function testing despite emerging evidence on the utility of preoperative platelet function testing in tailoring the timing of surgery [27] or predicting bleeding complications based on preoperative platelet function testing [28,29,30].

Platelet dysfunction is a major cause of bleeding following cardiac surgery [31,32].

Apparently, clinical trials are needed to assess the value of preoperative drug-specific platelet function testing in predicting bleeding complications.

Platelet function tests have the potential to predict bleeding complications and transfusion requirements; however, no “gold standard” test or cut-off value has yet been established.

Platelet function testing may be useful for monitoring platelet (dys) function during CPB and early after surgery upon arrival to the intensive care unit (ICU). Here, platelet dysfunction due to CPB and protamine may be of higher importance compared to platelet dysfunction caused by preoperatively administered APT. In different phases of cardiac surgery, platelet function is influenced by different modulating factors such as APT, CPB, and protamine administration for heparin reversal [14,15,33,34].

In fact, the multicausality of bleeding does not allow for strong correlations obtained by the assessment of a single hemostatic mechanism, and therefore, we must address some methodological considerations for studies investigating this issue. In this extremely challenging field, attention should be paid to every single detail and the appropriateness of study designs remains one of the most important issues. Compared to the other methodological issues in this field, the appropriate study design seems an easy reachable aim. Being aware that it is not possible to achieve an ideal experimental laboratory setting in real clinical life, we would like to address some very important methodological considerations for studies evaluating the prediction of excessive bleeding using HPOCT in cardiac surgery. Herein, we discuss methodological issues related to the investigation of the relationship between HPOCT and bleeding outcomes.

## 2. Methodological Considerations for Studies Evaluating the Prediction of Bleeding Complications Using Hemostatic Point-of-Care Tests in Cardiac Surgery

### 2.1. Type of Study

There are many different types of trials, and each type has distinct strengths and weaknesses (Figure 1). In general, there are two basic concepts that determine the type of study: (1) In regard to timing, studies may be conducted in a prospective or retrospective way. (2) In regard to patients’ exposure, studies may be either interventional or observational. Apparently, retrospective studies may only be observational, whereas prospective studies may be either interventional or observational in nature (Figure 1).

Different study settings were used in investigating the relationship between HPOCT and bleeding amount as well as transfusion outcomes [35].

### 2.2. Prospective Studies

From a historical viewpoint, the majority of studies pertaining to this issue were small prospective, observational cohort studies [35]. In recent times, there has been a growing proportion of retrospective studies [35]. When establishing research in this field, prospective observational studies are the most appropriate study type. In this way, it is possible to evaluate the association between observed parameters, i.e., platelet function test results and bleeding amount. The measurements may be performed at different time points and a correlation coefficient may be observed at each time point which reflects the different parameters influencing bleeding risk. For instance, performing platelet function testing prior to surgery may help to detect patients that have pronounced response to administered APT who could benefit either from earlier APT discontinuation or postponing surgery. At this time point, it is very important to use drug-specific platelet function tests that may direct further APT management. In cases of pronounced platelet inhibition, early discontinuation of APT should be considered. In contrast, if platelet function tests show high residual platelet reactivity despite administered APT, patients could undergo surgery even without APT discontinuation. In this way, it would be possible to avoid the rebound phenomenon which may occur after APT discontinuation in the subgroup of patients with hyperactive platelets [36].

Platelet function testing after weaning from CPB and heparin reversal by protamine may assess the effect of CPB and protamine on platelet function which may be particularly relevant in patients with long CPB times [33,34]. However, it is still a matter of debate whether platelet function tests or platelet contribution of clot firmness in viscoelastic testing better predicts bleeding complications in major cardiac surgery and other patients with thrombocytopenia [37].

However, prospective observational studies have some drawbacks that should be addressed. Even though consultant anesthesiologists and surgeons are unaware of the platelet function test findings, patients may receive procoagulant blood components according to predefined empirical clinical protocols. Thus, procoagulant blood component administration certainly hampers the observation of relationships between platelet function test results and the amount of postoperative bleeding by reducing the sensitivity of each test. It is not possible to assess the exact relationship between HPOCT results and the amount of bleeding, as the latter is the target of treatment, thus influenced by anesthetic measures during surgery. Obviously, in clinical situations it would not be ethically acceptable to avoid administration of procoagulant blood component therapy in bleeding patients, thereby making this methodology be compromised. Furthermore, prospective studies usually recruit only a relatively small number of patients which makes them frequently underpowered for assessment of some important clinical outcomes (i.e., mortality) which would confirm the relevance of bleeding complications.

If prospective observational studies detect a significant association between platelet function test values and bleeding extent, receiver operating characteristics (ROC) curve analysis should be performed [38] with the aim of delineating the cut-off value with the best sensitivity and specificity in predicting a particular outcome such as excessive bleeding. ROC curve analysis may define cut-offs that have the best positive predictive values (PPV) and negative predictive values (NPV) [38]. ROC curve analysis-derived cut-off values may direct the clinical decision-making process for preoperative APT discontinuation management or intraoperative transfusion management. The definition of cut-off values that delineate bleeding tendency presents the cornerstone according to which interventions may be directed. Interventions based on ROC curve analysis cut-offs may direct clinical practice, and may be used as a subject to further evaluate through interventional studies.

Prospective studies in this field should be designed as a part of a research project encompassing three stages: (1) Monocentric prospective studies evaluating the association between hemostatic blood properties and bleeding complications. In this stage, cut-off values that delineate bleeding complications should be defined. (2) Multicentric validation studies should confirm the association and cut-off values detected in the first step. (3) Prospective interventional studies should confirm the causal relationship and usefulness to guide interventions to prevent or stop bleeding and to develop validated hemostatic algorithms based on pre-defined cut-off values as recommended by the bleeding management guidelines [25].

There is a shortage of prospective interventional studies in this field. Mahla et al. conducted the first prospective interventional study using a platelet function testing-based strategy to reduce the bleeding and waiting time in clopidogrel treated patients undergoing coronary artery bypass grafting (CABG) [27]. In the absence of validated cut-off values that predict excessive bleeding, authors divided patients into three groups according to expert opinion-based cut-off values that might delineate bleeding tendency [27]. Even though this study used surrogate cut-off values, the authors reported positive results with a reduction in surgery waiting time after clopidogrel withdrawal without an increased risk of excessive bleeding [27]. This study shows how the definition of cut-off values can be performed through interventional studies. In cases of preoperatively performed platelet function tests, the intervention may be either the decision to discontinue or continue with APT preoperatively or to guide the timing of surgery based on pre-defined cut-off values [27].

Weber et al. [39] conducted a prospective randomized clinical trial aiming to evaluate the efficacy of hemostatic therapy guided by an HPOCT algorithm vs. an algorithm guided by conventional coagulation tests in patients undergoing major cardiac surgery experiencing clinically relevant bleeding after weaning from CPB and heparin reversal by protamine [39]. Algorithms for hemostatic management were based on either conventional coagulation tests or HPOCT (thromboelastometry and whole blood impedance aggregometry) and covered intraoperative as well as postoperative hemostatic management [39]. The study was terminated early since the planned interim analysis showed a highly significant reduction in RBC transfusion with a *p*-value < 0.01 according to predefined conditions. Furthermore, according to references [40,41,42,43], the patients in the HPOCT group showed a significant reduction in 6-month mortality (4% vs. 20%; *p* = 0.013) [39]. To the best of our knowledge, this is the first prospective randomized controlled study that showed clinical benefits when intra- and postoperative hemostatic management was based on HPOCT [39]. However, a significant improvement in patient outcomes and hospital costs could be confirmed in a multicenter trial looking at 5-year mortality, particularly in patients with long CPB times (>115 min) and bleeding after weaning from CPB [44].

### 2.3. Retrospective Studies

In recent years, there has been a growing number of retrospective cohort studies [40,41,42,43]. Authors may often consider these studies as retrospective analyses of prospectively collected data. This essentially means that authors have created robust databases with all the relevant data including the data on platelet function and viscoelastic testing, bleeding complications, and transfusion requirements. However, retrospective studies are the least appropriate study type for bleeding risk assessment based on HPOCT. In such studies, HPOCT results are usually used in regular clinical practice as a part of clinical decision-making. This results in major bias since the results of platelet function and viscoelastic testing were used as a part of hemostatic management, and therefore, may have influenced both the amount of bleeding and transfusion requirements as a self-full-filling prophecy. However, such a concept of retrospective analysis of prospectively collected data may provide valuable and useful data, but the observed associations should always be interpreted carefully due to the inextricable presence of bias. Ranucci et al. reported a retrospective analysis of prospectively collected data aiming to assess the relationship between platelet function testing and the amount of CTD [40]. Here, it is important to stress that patients with observed microvascular bleeding were treated with desmopressin if the preoperative whole blood impedance aggregometry (Multiplate) ADP test was less than 40 AUC [40]. Apparently, retrospective studies have some obvious shortcomings that actually lead to cut-off values which may be distorted with concomitant use of observed parameters in regular clinical practice [40]. Apart from these shortcomings, these studies may have some advantages [40]. Since they usually rely on huge databases, specific subgroup analyses are feasible which in turn may provide useful and clinically relevant data that would otherwise be time-consuming to obtain throughout prospective observational studies [42]. Furthermore, this type of study is inexpensive and provides results quickly once a database is created. The presence of confounders must be considered, even in cases where the propensity score matching the data analysis is used because such analysis may control only for known cofounders that are available in the database.

### 2.4. Definition of Bleeding Outcomes

For cardiac surgery patients, abnormal postoperative bleeding remains difficult to define. There is no generally accepted definition of excessive postoperative bleeding in cardiac surgery patients (Table 1). Although some authors offer definitions of abnormal blood loss [23], there is no consensus on the definition of excessive bleeding. Furthermore, the definition of excessive bleeding may be dependent on the kind of cardiac surgery such as routine CABG, valve replacement, complex cardiac surgery, or lung and heart transplantation. Different bleeding classifications exist but are not easily applicable to cardiac surgery patients [45]. Ti et al. discussed this issue [23]. Blood loss, measured as CTD, is a continuous parameter; however, a separation of normal and abnormal bleeding based on a single numerical value is necessary to allow meaningful conclusions [23]. Furthermore, CTD consists of a mixture of fluids, including actual blood loss, serous drainage, and fluid left in the pleural cavity. Furthermore, the actual blood loss through these tubes is the sum of bleeding due to hemostatic disorder and bleeding from wound edges due to surgical issues. When conducting research evaluating the relationship between HPOCT and bleeding outcomes possibility of surgical bleeding should inevitably be considered. Surgical bleeding affects to some degree the extent of postoperative CTD. In 70% of patients undergoing re-exploration for excessive bleeding, a surgical cause of bleeding (bleeding vessels and anastomoses) has been identified [46]. Such patients should be excluded from the analysis. The study’s participants should be well-balanced in regard to surgeons performing hemostasis because there is evidence that individual surgeons’ performance may contribute significantly to bleeding outcomes [46]. Meticulous surgical hemostasis should be the cornerstone of appropriate hemostatic management as it influences bleeding outcomes as much as preoperative APT management and intraoperative transfusion management. For that reason, a formal operative checklist to reduce surgical bleeding should be part of the hemostatic protocol in every study evaluating the relationship between HPOCT and bleeding outcomes that aims to provide accurate and reliable correlations not distorted by factors unrelated to hemostatic alteration. Unfortunately, even then it is impossible to differentiate bleeding volume according to surgical or coagulopathic bleeding. The higher the proportion of “surgical bleeding” in CTD, the lesser the NPV for each HPOCT when predicting bleeding outcomes. Unfortunately, without surgical re-exploration, it is impossible to differentiate surgical bleeding and bleeding due to hemostatic disturbances. We suggest that re-exploration for excessive bleeding with bleeding vessels identified should inevitably be considered as exclusion criteria in this type of study.

International Initiative on Haemostasis Management in Cardiac Surgery provided a universal definition of perioperative bleeding (UDPB) in adult cardiac surgery [125]. Dyke et al. addressed the obvious lack of standardization in the definition of excessive bleeding [125]; with the attempt to precisely describe and quantify bleeding and to facilitate future investigations, the authors proposed a UDPB for adult cardiac surgery [125]. UDPB consists of five gradual bleeding categories based on several parameters such as re-exploration/tamponade, delayed sternal closure, the amount of CTD in the first 12 postoperative hours, transfusion of RBCs, FFP, PC, cryoprecipitate, prothrombin complex concentrate, as well as administration of recombinant factor VIIa [125]. Prior to this recent proposal for a standardized definition of excessive bleeding, published evidence lacked standardization in defining bleeding outcomes. Different definitions of bleeding outcomes exist (Table 1), and different definitions lead to different reported prevalence of excessive bleeding. Furthermore, heterogeneity in the definitions of bleeding outcomes certainly affects the ability of HPOCT to predict bleeding outcomes. Different definitions of bleeding outcomes result in different predictabilities of bleeding complications and pooling of the data from different studies is almost impossible in such a setting. We support the idea of a UDPB [125] even though one should be aware that the recently proposed definition by Dyke et al. [125] is based on factors that would be expected to be highly influenced by transfusion policies that are well known to vary widely among institutions and surgical teams and are prone to changes over time [21,102,125,126]. Despite all these possible shortcomings, the application of this definition in prospective studies may allow for the pooling of data from different studies. Pooling of data may provide more reliable data on the role of HPOCT in the prediction of bleeding complications.

The same holds true for transfusion requirements. Thresholds for transfusion of blood components vary widely among different centers and sometimes are based on anesthesiologists/surgeons personal experiences and preferences. The exact thresholds for transfusion of each blood component and other hemostatic therapy should be defined and, if possible, used consistently in studies that aim to provide the pooling of the evidence that would allow for meaningful conclusions. Therefore, the monitoring of adherence to the protocol (algorithm) is of outstanding importance in prospective interventional studies and should always be reported in order to avoid misinterpretation of study results [127,128,129]. Apparently, to extract the best out of the standardized definition, each component that contributes to the definition should be standardized, which is particularly important in transfusion management.

For research centers that, for some reason, consider Dyke’s definition of perioperative bleeding not to be suitable or feasible, we propose to create their definition in order to adjust the CTD volume to anthropometric properties of their cohort, as well as to surgical technique, perfusion, and anesthetic management. As stressed earlier, the separation of normal and abnormal bleeding based on a single numerical value is necessary, and, herein, we propose our way to define excessive bleeding. The CTD amount should inevitably be divided by the patient’s weight and the patients should be characterized as bleeders if their 24-h CTD (mL/kg) exceeds the 75th percentile of distribution within the study cohort. This way of excessive bleeding definition has already been described in the literature [29,30,74]. We believe that such a definition allows for the most reliable correlation, and is not distorted with different perfusion, surgical, and anesthesia techniques attributable to different centers. The amount of postoperative bleeding may be recorded in different timeframes, and measurement of the CTD on an hourly basis allows for precise bleeding dynamic assessment. Assessment of bleeding amounts in an hourly manner may add some information on the predictability of bleeding in early postoperative hours versus bleeding in the late postoperative phase.

Considering the prediction of bleeding outcomes using HPOCT, the majority of studies report poor PPVs, whereas NPVs are quite high [35]. The question of how to improve PPVs remains challenging. One of the parameters that certainly influence PPV is the transfusion of procoagulant blood components in the intra- and postoperative phases. For instance, in our previously published prospective observational studies [29,30], clinicians were not aware of the HPOCT results. However, they managed patients according to the best available hemostatic management and patients were transfused with procoagulant blood components, whenever deemed necessary. Thus, patients with poor HPOCT values may at the same time be treated with procoagulant therapy by attending the physician in line with his clinical judgment or regular clinical hemostatic management. In an observational study, such a transfusion may reduce the volume of CTD and consequently prevent bleeding complications (primary outcome) which in turn diminishes the PPV. Another reason for the low PPV of HPOCT is that microvascular bleeding is most often multifactorial and both, platelet function testing and viscoelastic testing, assess specific hemostasis issues only, and therefore, should be used complementarily.

When assessing bleeding complications, the amount of postoperative bleeding and transfusion requirements are inextricably associated. Rosengart et al. [43] evaluated the ability of preoperative platelet function testing to identify patients at increased risk of bleeding and transfusion outcomes. The authors described a very interesting way of defining bleeding outcomes in cardiac surgery patients [43]. CTD was dichotomized as normal versus significantly higher than normal [43]. Transfusion requirements were also coded dichotomously with regard to whether any amount of given product—allogeneic RBCs, fresh frozen plasma, or platelets concentrate—had been transfused intraoperatively or within 24 h postoperatively [43]. A composite outcome indicative of surgery-related bleeding events was defined as “high” CTD or procoagulant blood component transfusion [43]. In conducting research, such a composite endpoint may provide better positive predictive values as it incorporates two outcomes whose incidence is inversely related [43]. In this way, it is possible to control variables that hamper the evaluated correlations. In prospective observational studies, investigating the relationship between HPOCT and CTD amount, attending physicians are unaware of HPOCT results, thereby, administering procoagulant blood components according to regular clinical practice protocols. Transfusion of procoagulant blood components distorts correlations between the HPOCT results and bleeding outcomes in terms of reducing the sensitivity of HPOCT. Therefore, it is impossible to control transfusion effects on bleeding outcomes, and the use of composite endpoints including both, bleeding amount and transfusion of procoagulant blood components, sounds reasonable.

### 2.5. Bleeding Risk Stratification Models

Identification of patients who are at increased bleeding risk would be advantageous in optimizing perioperative management.

Because of multicausality, it is a challenge to develop a bleeding risk prediction model that would be reliable and reproducible, worldwide.

Vuylsteke et al. developed a bleeding risk stratification score to identify patients undergoing cardiac surgery with a higher risk of severe postoperative bleeding [130]. Excessive bleeding was defined as blood loss exceeding 2 mL/kg/h during the first three hours following admission to the ICU [130]. A risk stratification score was constructed to define three categories: (a) high, (b) medium, and (c) low risk of severe postoperative bleeding with rates of severe postoperative bleeding being 21%, 8%, and 3%, respectively [130]. Using this bleeding score [130], authors were able to separate preoperatively patients into bleeding risk groups.

Greiff et al. conducted a study to perform external validation of the Papworth Bleeding Risk Score and to compare the usefulness of the UDPB, proposed by Dyke et al., with the Papworth Bleeding Risk Score [102]. In addition, the authors developed their local bleeding prediction score using stricter statistical criteria. When those two prediction models were compared, both showed a similar pattern with a high NPV but very low PPV, indicating that both scores were superior in identifying patients at low, rather than high risk of bleeding. Even though UDPB seems to be very attractive and calculates several parameters including transfusion requirement [125], it is strongly influenced by the transfusion practice of each institution. This is not the case for the definition of excessive bleeding proposed by Vuylsteke et al. [130]. However, in the study published by Greiff et al. [102], the application of either definition of excessive bleeding did not obtain significantly different results. Therefore, it seems that there is room for improvement and refinement in the development of bleeding stratification models. Even though the present bleeding stratification models do account for preoperative APT administration, they do not account for the platelet inhibitory effect that these drugs reveal. Furthermore, several other drugs not considered “antiplatelet drugs” but often used in cardiac and critically ill patients are known to impact platelet function such as protamine, beta-blockers, calcium antagonists, beta-lactam antibiotics, anti-depressants, and analgetics [131]. We hypothesized that the addition of HPOCT results may improve the PPV of bleeding stratification models, which was found to be the major weakness of the present models [102,130]. Our research group recently developed a SHOULD-NOT-BLEED Score [132]. To address the issue of unnecessary transfusions within the CABG population, we developed a model to predict which patients are at low risk of bleeding for whom transfusion treatment might be considered unnecessary [132]. The study aimed to develop a user-friendly application that stratifies patients with respect to bleeding risk. The statistical model we used to develop the SHOULD-NOT-BLEED Score aimed to detect CABG patients at low risk of bleeding [132]. We developed a Windows platform app based on risk modeling which we previously calculated for 1426 patients undergoing elective CABG [133]. The variables that entered the scoring system were as follows: Age; Body Mass Index; Chronic renal failure; Preoperative clopidogrel exposure; Preoperative red blood cell count; Preoperative fibrinogen level; Preoperative multiplate ASPI test area under the curve (AUC) units [132,133]. The score included a platelet function testing variable and predicted patients without a risk of excessive bleeding with strong discriminatory performance (Receiver Operating Curve (ROC) analysis AUC 72.3%, *p* < 0.001) [132,133].

### 2.6. Patient Selection

Selected patient cohorts should be homogeneous as much as possible since patients’ heterogeneity in terms of patient comorbidities, complexity of procedures, and preoperative APT management may diminish the predictability of HPOCT in the assessment of bleeding outcomes. When considering patients to be recruited in studies evaluating the relationship between platelet function testing and bleeding outcomes, one should be aware that with increasing bleeding risk, the higher the possible benefit of the bleeding risk assessment using HPOCT. Therefore, research should be focused on patients considered to be at high bleeding risk.

Preoperative APT management should be consistent among study participants. It seems that preoperative platelet function testing is of limited value in predicting bleeding risk in patients not being exposed to APT, preoperatively [35]. Recently, our working group has published a systematic review evaluating the predictive value of POC PFT for postoperative blood loss and transfusion requirements in cardiac surgery [35]. In a negligible proportion of reviewed studies, patients were not exposed to APT, preoperatively [35]. Moreover, almost half of the studies enrolled patients undergoing isolated CABG, even without the use of CPB [35]. Furthermore, it seems that the great majority of patients actually underwent elective surgery procedures [35]. The results from these studies may only be reliably applied to similar patient populations. Based on the reviewed literature, this means that we should draw the conclusions only from patients undergoing isolated CABG with APT not being administered, preoperatively. In the new era of cardiac surgery, with a growing proportion of high-risk patients undergoing complex procedures with recent exposure to APT, it seems reasonable to shift our focus to this subgroup of patients, as they may benefit the most from further refinements in hemostatic management. Despite the current guidelines, many centers resume with APT until the day of surgery, disregarding the recommendations for a drug-free interval before surgery. For example, the data from 2858 acute coronary syndrome patients in the CRUSADE (Can Rapid risk stratification of Unstable angina patients Supress Adverse Outcomes with Early implementation of the ACC/AHA Guidelines) initiative demonstrated that 87% of patients treated with clopidogrel preoperatively underwent CABG surgery ≤ 5 days after treatment cessation with the result of increased blood transfusion requirements [134]. Furthermore, 5–15% of patients with acute coronary syndrome require urgent cardiac surgery with recently administered APT [134]. These real-life data should direct our research efforts to the subgroup of patients that could benefit the most from improved hemostatic management.

Furthermore, patients exposed to more potent APT should be the focus of research. Studies should be conducted to elucidate the pharmacodynamic profile of novel APT. More pharmacodynamic studies of platelet function recovery after discontinuation of prasugrel and ticagrelor should be conducted. A platelet inhibitory response as well as the recovery rate should be calculated using drug-specific platelet function tests. It is very important to compare these characteristics in cardiac surgery patients as this may influence APT administration/discontinuation management. Another important question is whether the uniform cut-off levels of adenosine diphosphate (ADP) platelet reactivity reflect bleeding tendency. If this hypothesis were confirmed, one could assess the platelet inhibitory response to each ADP platelet receptor blocker and create a unique ADP test cut-off value that corresponds to the bleeding risk. This would allow for the detection of patients with a high response to clopidogrel therapy and these patients may have a similar ADP platelet inhibition as patients exposed to more potent ADP receptor blockers. One could assume that a similar level of platelet ADP receptor inhibition may be achieved with clopidogrel, prasugrel, and ticagrelor but with different probability odds to achieve certain levels of platelet reactivity. More potent ADP receptor blockers may have a higher odds to achieve pronounced platelet inhibition whereas less potent ADP receptor blockers (such as clopidogrel) may have a higher odds for high residual platelet reactivity. These pharmacodynamics hypotheses should be tested in upcoming studies. However, one should be aware that not one single type of receptor activity reflects overall platelet reactivity. In patients treated with ADP receptor blockers, a protease-activated receptor (PAR) that reacts with thrombin stimulation may still be active. Recently, Ranucci et al. evaluated the effect of preoperative ADP and thrombin receptor inhibition on bleeding after cardiac surgery [110]. Both ADP and thrombin receptor-activating peptide (TRAP) test, were significantly associated (*p* < 0.001) with postoperative bleeding. However, weak ADP receptor activity, if coupled with sufficient TRAP test was not associated with severe bleeding [110]. It turns out that overall platelet reactivity may remain within an acceptable range even in the presence of strong drug-induced inhibition of ADP receptors, which suggests a compensatory mechanism [110]. Even though they were obtained in different clinical settings of percutaneous coronary intervention, a very similar phenomenon has been described by Tricoci et al. [135]. In addition, the role of aspirin should not be underestimated as there is evidence that high platelet inhibition measured in patients while on aspirin therapy (aspirin hyperresponse) may contribute to bleeding diathesis [30]. Accordingly, the take-home message would be that not only one drug-specific test but rather a combination of specific tests should be performed at the same time. Based on recently published data, it seems that impaired the TRAP test should be used to justify platelet transfusion due to low ADP test results [110]. In this way, concomitant assessment of different platelet activation pathways (arachidonic acid, ADP, and TRAP) may reliably provide a level of general platelet reactivity. This strategy detects patients who may proceed with surgery even if they are exposed to ADP receptor blockers and have weak ADP receptor activity [110]. It is well known that a platelet inhibitory response to APT varies widely among individuals [136] and such considerable heterogeneity in platelet inhibitory response to APT makes it difficult to reliably use arbitrary intervals for drug discontinuation prior to surgery without incurring excessive thrombotic or bleeding risks with premature versus too late discontinuation. Therefore, platelet function testing seems reasonable in a group of patients preoperatively exposed to APT in order to detect patients with pronounced platelet inhibition who might benefit from earlier drug discontinuation or delaying surgery until platelet recovery, if clinical condition allows. Vice versa, continuing APT up to the day of surgery in patients who have high residual platelet reactivity may help to avoid adverse thrombotic events associated with platelet recovery and possible rebound phenomena. Regarding preoperative APT management and patient selection, research should be focused on and provide priority to patients exposed to dual APT in close proximity to surgery. Furthermore, special attention should be paid to patients exposed to more potent APT such as prasugrel [121,137,138] and ticagrelor [138,139]. Pharmacodynamic evaluation of novel, more potent APT and the clinical implications of switching between them in the preoperative phase should be considered in future trials.

Furthermore, patients should be comparable in regard to the complexity of the surgical procedures performed. First, recruited patients should be equal in regard to the use of CPB during the surgical procedure. Merging patients undergoing off-pump and on-pump cardiosurgical procedures would be detrimental in terms of creating great heterogeneity and bias. More complex procedures are associated with a higher bleeding risk attributable to surgical complexity and duration of CPB. In addition to complexity and length of incision-to-suture times, there are several other factors related to CPB that contribute to the onset of hemostatic disorder such as foreign surface contact, consumption of fibrinogen and enzymatic coagulation factors, platelet dysfunction, fibrinolysis, hypothermia, acidosis, and hypocalcemia [11]. The multicausality of hemostatic disorders related to CPB requires well-balanced homogeneous populations with a minimum risk for confounding factors since heterogeneity of patient population makes it difficult to exclude the effects of complexity of the cardiac pathology condition, such as levels of hypothermia and duration of CPB, on the postoperative blood loss.

Considering patient selection, we would like to provide case-based learning. Agarwal et al. [112] recently conducted a prospective randomized control trial aiming to investigate whether the use of preoperative platelet function testing as a part of a transfusion algorithm may reduce transfusion requirements in cardiac surgery patients [112]. In our opinion, patient selection remains the main shortcoming of the study [112]. This study enrolled both patients undergoing elective and urgent surgery. Furthermore, they enrolled patients who underwent various types of cardiosurgical procedures and cohorts consisted of both, patients undergoing cardiac surgery with and without CPB [112]. Accordingly, we may conclude that the heterogeneity of the study cohorts makes it difficult to control the confounding factors [112]. Even if studies report positive results, as in this study [112], the observed results may be distorted and hampered by the heterogeneity of the study cohorts [112]. With this example, we would like to stress how valuable and robust studies may be improved by addressing some methodological considerations. In our opinion, such a type of study should have a narrow focus on one particular subgroup of cardiac surgery patients. The first discrimination should be with respect to the use of CPB and the second discrimination should address different types of procedures with respect to surgical complexity. Surgical complexity certainly affects bleeding outcomes and transfusion requirements due to the extent of surgical stress with the number of suture lines and the duration of CPB time which has deleterious effects on hemostasis, particularly in patients with a long CPB time.

Accordingly, the utility of HPOCT-guided hemostatic management should be the highest in the group of patients considered to be at the highest bleeding risk, which in turn may be determined with preoperative exposure to APT as well as with complexity of procedure (long CPB time with hypothermia, sometimes deep hypothermic circulatory arrest).

### 2.7. How to Set Up Appropriate Timing for Point-of-Care Measurements

Because the platelet dysfunction may be caused by either APT being administered preoperatively or by CPB effects, it is necessary to make a clear distinction between these two potential causes of platelet dysfunction. However, in procedures with long CPB times, both causes may contribute to postoperative platelet dysfunction. To make a clear distinction between these two causes, it is necessary to study patients undergoing CPB-assisted cardiac surgery procedures with preoperative exposure to APT. In this type of trial, platelet function assessment should be performed at different time points. Thereby, we would be able to reliably describe the mechanisms by which APT and CPB achieve harmful effects of platelet function. This should be coupled with assessing the clinical relevance of the described phenomenon by correlating the hemostatic properties (i.e., platelet function) with relevant clinical outcomes such as bleeding amount or transfusion requirements.

When considering hemostatic properties in cardiac surgery patients, clinicians are usually focused on the following:(1)Baseline (preoperative) hemostatic properties, in particular platelet function that may often be under considerable influence of preoperatively administered APT;(2)Intraoperative changes in hemostatic blood properties that are mainly influenced by the effects of CPB or heparin reversal by protamine;(3)Hemostatic properties following arrival to the ICU is extremely important and should not be underestimated [1].

Hemostatic alterations following arrival to the ICU are related to hemostatic disorder secondary to blood loss and dilution followed by fibrinolysis. In particular, attention should be paid to fibrinolytic bleeding, especially in centers that perform reinfusion of mediastinal blood (without washing) which in turn can result in systematic fibrinolytic coagulopathy [140] and centers that do not administer antifibrinolytic therapy topically or systemically during surgery [141,142].

Hemostatic alterations in cardiac surgery patients are moving targets, constantly changing over time during and after CPB and following admission to the ICU. Appropriate timing for HPOCT is of utmost importance. In general, the more frequent the timing for measurements, the better the approach to detecting hemostatic alterations that rapidly change over time.

The evaluation of the relationship between CPB duration and hemostatic disturbances is very important in the assessment of bleeding risk. Recently, Mutlak et al. [143] conducted a prospective observational study that aimed to assess the capability of the whole blood impedance aggregometry (Multiplate) to assess the extent of CPB-associated platelet dysfunction [143]. They showed that platelet function, assessed by ADP test reflected the time-dependent extent of CPB-induced platelet dysfunction [143]. The authors provided valuable data on the dynamics of platelet function during CPB; however, failed to correlate these findings with clinical outcomes. In our opinion, such studies should be focused on the gaps in our knowledge regarding this specific issue. The observed phenomena, such as the dynamics of platelet function over time on CPB, should inevitably be analyzed in the context of clinical outcomes such as bleeding complications and transfusion requirements. Accordingly, the evaluation of platelet function should inextricably be coupled with clinical endpoints, in this type of trial. Only in this way, it will be possible to draw conclusions that are of clinical relevance.

HPOCT may help in clarifying the mechanisms of hemostatic disorders before, during, and after CPB. Based on our own experience [28,29,30,144], we may conclude that PFTs may be more useful in managing platelet dysfunction caused by exposure to APT. Using these devices, it is possible to perform bleeding risk stratification by detecting patients with pronounced platelet inhibition (hyperresponders) [30]. Furthermore, PFTs may be useful for the assessment of platelet dysfunction during and after CPB, although it seems that viscoelastic coagulation tests are of greater usefulness in terms of diagnosing mechanisms for hemostatic disorder as well as directing transfusion practice [29,39,145].

Presumably, the assessment of platelet function at different time points may elucidate the relative role of each modulating factor. Later measurement time points may more accurately predict bleeding as they assess platelet function influenced by accumulative effects of different modulating effects such as APT, CPB, and protamine. However, the platelet function assessment in earlier periods (i.e., in the preoperative phase) enables a wider spectrum of hemostatic measures to prevent bleeding complications such as personalized timing of surgery or personalized APT administration/discontinuation management.

### 2.8. Data Analysis—Statistical Considerations

#### Sample Size Calculation

A minimum sample size required for a study is a major consideration which all scientists need to address at an early stage of creating a research protocol [146]. Furthermore, for type I errors and power of the study, some estimates for the effect sizes will also need to be determined in the process of calculating or estimating the sample size [146]. The appropriateness of calculating or estimating the sample size will enable the researchers to better plan their study, particularly pertaining to the recruitment of subjects and to avoid underpowering the study [146]. The type of sample size calculation depends on the study design and the type of statistical analyses performed to compare subjects and/or outcomes.

### 2.9. Which Tests Should Be Used and When?

#### 2.9.1. Correlation Tests

Studies evaluating the association of bleeding outcomes with haemostatic properties or subject properties usually rely on correlation tests. Correlation is a bivariate analysis that measures both the strength and direction of association between two variables. Regarding the strength of the relationship, the value of the correlation coefficient varies between +1 and −1, where a value of ±1 indicates a perfect degree of association between the two variables. As the correlation coefficient value goes towards zero, the relationship between the two variables will be weaker. The direction of the relationship is indicated by the sign of the coefficient; a + sign indicates a positive relationship and a negative sign indicates a negative relationship. In statistics, there are four types of correlations: Pearson correlation; Kendall rank correlation; Spearman correlation; and the Point-Biserial correlation.

Pearson r correlation is the most widely used correlation statistic to measure the degree of the relationship between linearly related variables. For the Pearson r correlation, both variables should be normally distributed (normally distributed variables have a bell-shaped curve). Spearman rank correlation is a non-parametric test that is used to measure the degree of association between two variables. The Spearman rank correlation test does not carry any assumptions about the distribution of the data and is the appropriate correlation analysis when the variables are measured on a scale that is at least ordinal.

#### 2.9.2. ROC Curve Analysis

Using diagnostic testing to determine the presence of a specific outcome (be it the presence or absence of) i.e., excessive bleeding is essential in clinical research [147]. In many cases, test results are obtained as continuous values and require a process of conversion and interpretation into a dichotomous form to determine the presence of a disease or outcome [147]. The primary method used for this process is the receiver operating characteristic (ROC) curve analysis [147]. The ROC curve analysis is used to assess the overall diagnostic performance of a test and to compare the performance of two or more diagnostic tests [147]. It is also used to select an optimal cut-off value for determining the presence or absence of a disease or outcome [147]. Furthermore, for optimum cut-off values, ROC curve analyses provide information about accuracy (ROC AUC), significance (standard error, 95% confidence interval, and *p*-value), sensitivity, specificity, positive predictive value, and negative predictive value.

### 2.10. Sensitivity, Specificity, Positive Predictive Value, and Negative Predictive Value

Monaghan et al. recently described foundational statistical principles in medical research such as sensitivity, specificity, positive predictive value, and negative predictive value [148].

Briefly, sensitivity denotes the proportion of subjects correctly given a positive assignment out of all subjects who are actually positive for the outcome. On the other hand, specificity denotes the proportion of subjects correctly given a negative assignment out of all subjects who are actually negative for the outcome [148]. A positive predictive value reflects the proportion of subjects with a positive test result who truly have the outcome of interest whereas a negative predictive value reflects the proportion of subjects with a negative test result who truly do not have the outcome of interest. According to ref. [148], sensitivity and specificity are inversely related, wherein one increases as the other decreases. Positive and negative predictive values do inherently vary with pre-test probability (e.g., changes in population disease prevalence). Positive and negative predictive values are influenced by the prevalence of disease in the population that is being tested. If we test in a high prevalence setting, it is more likely that persons who test positive truly have the disease than if the test is performed in a population with low prevalence [148].

## 3. Conclusions

The developed models for prediction of excessive bleeding show consistently low PPVs coupled with acceptable to high NPVs indicating that available scores are better in identifying patients at low risk rather than high risk of bleeding [102]. Accordingly, it seems easier to identify patients who should not bleed rather than patients who should bleed. This may have numerous implications for clinical practice [102] in terms of selecting patients for whom HPOCT might not be necessary.

For studies addressing this research field it is of utmost importance to adequately select patients (i.e., patients exposed to APT preoperatively who require complex surgical procedures using CPB). Particular attention should be given to patients with renal failure.

Preoperative APT certainly influences bleeding outcomes. Well-known wide variability in the platelet inhibitory response to APT directs how powerful a correlation will be between APT and bleeding outcomes. Inclusion of PFT should inevitably be considered for bleeding risk stratification as PFT may clearly distinguish patients with pronounced platelet inhibition (hyperresponders) with a high bleeding risk, from those with high residual platelet reactivity (low responders) who may in fact proceed to surgery without risk of bleeding despite APT administered in close proximity to surgery. Appropriate methodology in conducting such studies may be of utmost importance leading to adequate bias control.

Even if positive correlations between preoperative HPOCT and bleeding outcomes are obtained through well-designed prospective studies, it may be too optimistic to expect that only one PFT assay may be sufficient to predict bleeding. The adequate value of a single patient characteristic or a single laboratory biomarker in the prediction of excessive bleeding has not been proven, yet [149,150]. The best bleeding risk prediction score should consider non-surgical risk factors (age, sex, renal disease, administration of antiplatelet/antithrombotic drugs, platelet function test values) and surgical procedure-related risk factors. In order to optimize perioperative management, bleeding risk prediction scores should provide better PPVs. With acceptable PPVs, it would be possible to control and treat factors contributing to the bleeding risk.

It is obvious that there is a need to consider the diagnostic approach and the management of hemorrhage in cardiac surgery. Further refinements in hemostatic management are warranted. Newly designed hemostatic algorithms should inevitably encompass preoperative, intraoperative, and postoperative time points.

The preoperative part should provide a preoperative bleeding risk stratification based on the assessment of platelet function in relation to APT administration/discontinuation management. Exact cut-off values that delineate the bleeding risk should be defined using drug-specific platelet function tests, and preoperative APT administration/discontinuation management should be adjusted, accordingly.

Considering the intraoperative part, again, exact cut-off values that delineate excessive bleeding should be defined and validated. Assessment of viscoelastic blood clot properties provides greater usefulness to diagnose bleeding mechanisms and should be used complementary to PFT in bleeding patients. Platelet function testing appears to be more useful for the preoperative assessment of platelet function in patients being exposed to APT, preoperatively. Notably, some authors reported intraoperative administration of desmopressin or even platelet concentrates based on platelet function testing results performed intraoperatively in cases of microvascular bleeding occurrence [40]. However, here, viscoelastic testing seems to be more useful and reliable.

Postoperative hemostatic measures should provide a differential diagnosis of the underlying hemostatic disorders, such as dilutional coagulopathy, fibrinolysis, platelet dysfunction, or fibrinogen deficiency. On-time detection of surgical bleeding should be given priority. Any assessment of possible hemostatic disturbances starts with two questions and answers:Question 1: Do we have wet surgical field?—Answer 1: YesQuestion 2: Do we have surgical bleeding?—Answer 2: No

The use of HPOCT makes no sense in cases of the dry surgical field. If surgical bleeding persists, HPOCT should generally not be used as an alternative to appropriate surgical haemostasis. In general, if bleeding occurs in patients considered to be at low risk of bleeding, the focus should be directed toward possible surgical causes for excessive bleeding. However, in real clinical scenarios, it is not always “either black or white” but rather “shades of gray” scenarios where we may have “small” surgical sources of bleeding that could be stopped if the patient’s haemostatic properties were optimized. Patients that are primarily considered to be at low risk of bleeding may undergo a procedure with, for example, a long pump run time, which in turn may significantly alter the haemostasis. With such haemostatic disturbances patient may experience “oozing” and a wet surgical field resulting in excessive chest tube output. In such cases, HPOCT-guided optimization of the haemostatic blood properties may be considered as a first step to stop bleeding, and if bleeding persists despite optimized haemostasis, surgical re-exploration may be the next step. The message herein is that there are some clinical scenarios where we may experience surgical bleeding with small sources of bleeding that could efficiently be treated with optimizing blood haemostasis rather than immediate chest re-exploration. Such an approach may optimize clinical outcomes by avoiding unnecessary surgical revisions.

## Figures and Tables

**Figure 1 jcm-13-06737-f001:**
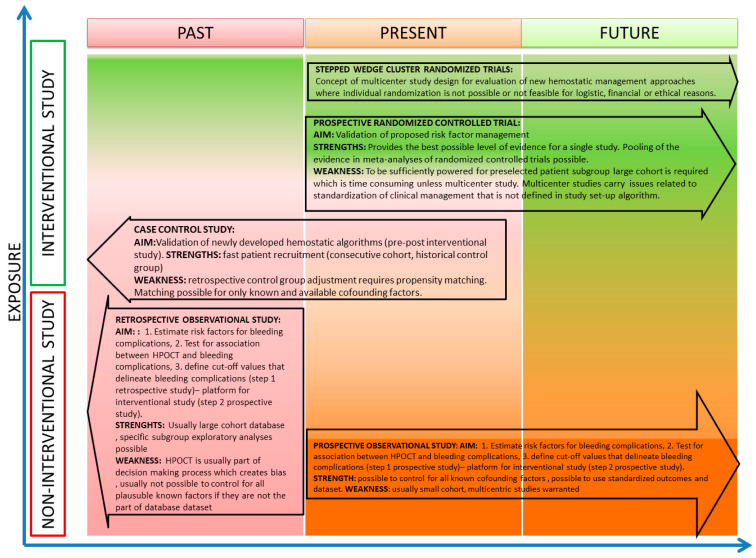
Methodological considerations for studies with respect to intervention and timeline.

**Table 1 jcm-13-06737-t001:** Various definitions of excessive bleeding across studies recruiting patients undergoing cardiac surgery procedures.

Author	Year of Publication	PMID	Definition of Excessive Bleeding
Marengo-Rowe et al. [47]	1979	382479	In the postoperative period, excessive hemorrhage was defined as that exceeding 600 mL CTD in the first eight hours.
Michelson et al. [48]	1980	6965824	When assessed as postoperative CTD volume, postoperative mediastinal bleeding of greater than 300 mL in the 1st hour, greater than 250 mL in the 2nd hour, and greater than 150 mL/h thereafter was regarded as excessive bleeding.
Bagge et al. [49]	1986	3738446	Heavy bleeding was defined as CTD greater than 800 mL/16 h.
Hirayama et al. [50]	1988	3406692	Excessive bleeding was defined as CTD exceeding 1 L.
Gram et al. [51]	1990	2114043	Excessive bleeding was defined as CTD exceeding 520 mL.
Ratnatunga et al. [52]	1991	1863147	Excessive bleeding was defined as CTD exceeding 400 mL in the first postoperative hour, or more than 300 mL in each of the first 2 h, or more than 200 mL in each of the first 3 h, or more than 100 mL in each of the first 4 h.
Villarino et al. [53]	1992	1375613	Excessive bleeding was defined as CTD of greater than or equal to 1000 mL within 4 h of surgery.
Wang et al. [54]	1992	1540061	Excessive bleeding was defined as CTD > 10 mL/kg for the first hour and more than 200 mL/h during the first 6 h after operation
Hartstein et al. [55]	1996	8957439	Excessive bleeding was defined as CTD > 250 to 300 mL/h for the first 2 h followed by >150 mL/h thereafter.
Despotis et al. [56]	1996	8712388	Excessive bleeding was defined using the following four criteria: (1) >100 mL in any postoperative hour; (2) >150 mL in any postoperative hour; (3) >1000 mL; or (4) >1600 mL cumulative CTD in the first 24 postoperative hours.
Despotis et al. [57]	1996	8968178	Excessive microvascular bleeding was defined as diffuse bleeding from the surgical site without an identifiable surgical source.
Nuttal et al. [58]	1997	9412876	The anesthesiologist and surgeon evaluated blood loss 10 min after protamine administration. The patient was characterized as a “bleeder” if both physicians determined the surgical field was “wet” (microvaseular bleeding).
Robert et al. [59]	1997	9356085	Excessive 24 h CTD was defined as losses greater than 1 L
Ereth et al. [60]	1997	9249097	More than 200 mL/h (or 100 mL/h) of CTD in the first 4 h in the ICU
Dacey et al. [61]	1998	9565127	Excessive bleeding was defined as bleeding that resulted in return to the operating room. No specific indication for re-explorations were reported. Decision to re-exploration was based on surgeons preference.
Ereth et al. [62]	1998	9579505	The two disease states of bleeding after CPB were determined by an average of 100 mL/h and 200 mL/h CTD in the first 4 h in the ICU.
Herwaldt et al. [63]	1998	9475343	The definition of hemorrhage following cardiothoracic operations required one of the following: reoperation for bleeding; postoperative loss of greater than 800 mL of blood over 4 h; or surgeon-diagnosed excessive intraoperative bleeding.
Despotis et al. [64]	1999	10408485	Excessive postoperative bleeding was defined as CTD > 1180 mL/24 h.
Mongan et al. [65]	1999	9706913	Excessive postoperative bleeding was defined as CTD > 1000 mL/24 h.
Lasne et al. [66]	2000	11127858	Mediastinal blood loss was recorded at 12 h. Calculation of blood loss was performed using Mercuriali’s formula. The patients were divided into two groups on the basis of blood loss values: below 926 mL, i.e., below the 75th percentile. Blood loss exceeding 926 mL (75th percentile) was considered as excessive bleeding.
			Latter group.
Casati et al. [67]	2000	10962414	Bleeding of more than 600 mL in the first 24 h was considered excessive postoperative bleeding.
Sachin et al. [68]	2001	11801823	Patients who bled > 100 mL/h for two consecutive hours were considered to have significant chest tube output.
Slaughter et al. [69]	2001	11302482	Excessive bleeding was regarded if 6 h mediastinal drainage was within the highest decile of cohort CTD distribution (exceeding 646 mL/6 h).
Ascione et al. [70]	2001	11279409	Excessive bleeding was defined as CTD > 150 mL/h over 2 consecutive hours.
Ereth et al. [71]	2001	11254840	Excessive bleeding after CPB was determined by greater than an average of 100 mL/h chest tube blood loss in the first 4 h in the intensive care unit.
Slaughter et al. [69]	2001	11302482	Cumulative 6 h CTD exceeding 646 mL/6 h.
Ti et al. [23]	2002	12407603	>1000 mL/24 h CTD or >250 mL for any two consecutive hours after arrival in the ICU.
Forestier et al. [72]	2002	12393769	CTD > 1 mL/kg/h for at least 1 h during the first 6 h after surgery.
Fattorutto et al. [73]	2003	12697601	Excessive CTD was defined as >200 mL for 2 successive hours.
Cammerer et al. [74]	2003	12505922	Abnormal bleeding: (1) CTD 750 mL/6 h; (2) CTD exceeding 75th percentile (500 mL/6 h postoperatively).
Casati et al. [75]	2004	15224025	Blood loss was recorded during the first 24 h, and excessive bleeding was defined as a blood loss greater than 600 mL in 24 h.
Pleym et al. [76]	2004	14980901	Excessive bleeding was regarded as CTD ≥ 200 mL/h or CTD ≥ 150 mL/h persisting for >3 h.
Poston et al. [77]	2005	15784355	24 h CTD > 800 mL.
Nuttal et al. [78]	2006	16551890	Excessive bleeding was defined by two criteria: (a) postoperating room chest tube blood loss over 24 h more than or equal to 750 mL (chest tube drainage [CTD] > or =750); and (b) transfusion of any non-red blood cell (RBC) blood products.
Carrol et al. [2]	2006	16581348	Bleeding assessment by observer agreement classified into the following categories: (1) not bleeding; (2) oozing; or (3) excessive bleeding.
Marietta et al. [79]	2006	16647479	Excessive bleeding was defined as >2 L chest tube output after surgery without pre-defined time frame for measurement of chest tube output.
Gerrah et al. [80]	2006	16868105	Severe blood loss > 965 mL of CTD.
Jimenez-Rivera et al. [81]	2007	17425777	Excehssive bleeding was defined as 24 h blood loss of >1 L post-CPB.
Yamada et al. [82]	2007	17458642	Excessive bleeding was defined as CTD greater than 2 mL/kg/h during the first 4 h after surgery.
Quattara et al. [83]	2007	17431000	Excessive bleeding was defined as chest tube output exceeding 500 mL during the first 24 h post surgery procedure.
Kim HJ et al. [84]	2008	19061702	Excessive bleeding was defined as a composite endpoint consisted of (1) packed red blood cell transfusion, (2) return to the operating room for bleeding, and (3) hematocrit drop of ≥15%.
Berger et al. [85]	2008	19007688	Major bleeding was defined as a >5 g/dL drop in hemoglobin, intracranial bleed, fatal bleed, or cardiac tamponade.
Davidson et al. [86]	2008	18922419	Bleeding of 200 mL or greater in a single hour was considered an abnormal result.
Reinhofer et al. [87]	2008	18388501	Excessive bleeding was defined as postoperative CTO ≥ 600 mL.
Rahe-Mmeyer et al. [88]	2009	19698858	After weaning from CPB, neutralization of heparin, and completion of surgical hemostasis and removing all blood from the wound area using suction device, dry wound area was thoroughly covered with sterile dry surgical swabs. The extent of blood loss was determined by measuring the difference in weight of the swabs before application and 5 min of adsorbing blood. Blood loss of 60 to 250 g was defined as high-level bleeding triggering coagulation therapy. Blood loss of greater than 250 g triggered additional surgical re-exploration.
Gill et al. [89]	2009	19546387	Excessive bleeding was defined as CTD ≥ 200 mL/h in any one hour after arrival to the ICU or ≥2 mL/kg/hr for two consecutive hours.
Preismann et al. [90]	2010	20181490	Cluster analysis revealed two groups of patients with respect to bleeding tendency. CTD was significantly higher in bleeding group (1216 ± 310 mL vs. 576 ± 105 mL).
Wasowicz et al. [91]	2010	20610554	Excessive blood loss was defined based on the number of post-CPB RBC transfusions. Specifically, it was defined as the transfusion of 5 U of RBCs from termination of CPB to 1 day after surgery.
Hermann et al. [92]	2010	20103308	Excessive bleeding was defined as presence of sings of tamponade or rexploration for excessive bleeding.
Nesher N et al. [93]	2010	20061339	Excessive blood loss was defined as >2 L of chest tube loss in the first 24 h, which corresponded to 2 S.D. above the mean for 24 h chest tube loss following isolated coronary surgery in author’s institution.
Kwak et al. [94]	2010	21126640	>200 mL/h in 2 consecutive hours.
Coakley et al. [95]	2011	21091865	Excessive bleeding was defined as the amount of CTD exceeding 1000 mL/24 h.
Ranucci et al. [40]	2011	21172499	≥800 mL/12 h CTD.
Mehran R et al. [45]	2011	21670242	Bleeding Academic Research Consortium defined CABG related excessive bleeding that included: (1) Perioperative intracranial bleeding within 48 h; (2) Reoperation after closure of sternotomy for the purpose of controlling bleeding; (3) Transfusion of ≥5 U whole blood or packed red blood cells within a 48 h period (only allogenic transfusions are considered transfusions for CABG-related bleeds); (4) Chest tube output ≥2 L within a 24 h period.
Weitzel et al. [96]	2012	22809250	>1000 mL/24 h CTD.
Lee et al. [97]	2012	22713683	CTD was stratified at the 75th and 90th percentile of CTD distribution in patient cohort. Bleeding outcome was dichotomized at 600 mL CTD (75th percentile) and 910 mL (90th percentile).
Deja et al. [98]	2012	22554721	Excessive bleeding was defined in two ways: (1) More than 750 mL of bleeding during the first postoperative 12 h; and (2) more than 1000 mL of total discharge from the chest drains.
Biancari et al. [46]	2012	22498634	Excessive bleeding requiring reexploration was noted in following conditions: (1) drainage > 500 mL during the first postoperative hour, >400 mL during each of the first 2 h, >300 mL during each of the first 3 h, or >1000 mL in total during the first 4 h; (2) continuous bleeding throughout the first 12 h, leading to total bleeding >100 mL/h; (3) sudden massive bleehding; (4) obvious signs of cardiac tamponade secondary to active or previous bleeding; (5) cardiac arrest of a patient who continued to bleed; and (6) excess bleeding despite the correction of coagulopathies.
Christensen et al. [6]	2012	22100857	Excessive bleeding was defined as postoperative drainage loss exceeding 200 mL/h in 1 h or 2 mL/kg for 2 consecutive hours occurring within 6 h after cardiac surgery.
Wang et al. [99]	2012	21737704	Excessive bleeding requiring re-exploration was considered when the chest tube drainage was >300 mL/h in the first 2 postoperative hours or >200 mL/h for 4 consecutive hours.
Biancari et al. [46]	2012	22498634	Postoperative blood loss was defined as the amount of blood loss from drainages measured on the morning of the first postoperative day or in the afternoon/evening in patients who underwent night time surgery. Postoperative blood loss was dichotomized according to 95th percentiles of postoperative blood loss (1600 mL).
Petricevic et al. [30]	2013	22926758	24 h CTD ≥ 11.33 mL/kg.
Petricevic et al. [29]	2013	23341179	24 h CTD ≥ 12.46 mL/kg.
Yang et al. [100]	2013	23710825	Clinically significant bleeding was prespecified as 3 mL/kg/h.
Emeklibas et al. [101]	2013	22934739	The 24 h chest tube blood volume of more than 1660 mL within 24 h after surgery (limit between third and upper quartile) was considered as excessive bleeding.
Ranucci et al. [7]	2013	23673069	Major bleeding was defined as blood loss (mL/12 h) greater than the tenth decile of the distribution or need for surgical revision owing to bleeding. According to the pre-stated definition, major bleeding was settled at the upper 10th decile of the distribution, sorrespondent to 900 mL/12 h.
Rosengart et al. [43]	2013	23953984	12 h CTD > 437 mL.
Greiff et al. [102]	2014	25529438	Treshold for excessive bleeding defined as ≥2 mL/kg/h within the first 4 hours postoperatively
Singh et al. [103]	2014	25392047	In observational study, the total mediastinal drainage ranged from 170 to 1200 mL with a mean of 52,564 ± 19,739 mL. The amount of >500 mL was defined as excessive bleeding determinant.
Welsh et al. [104]	2014	25239416	Significant bleeding was defined as sustained CTD of greater than 150 mL/h or more than 2 L/24 h.
Walden et al. [105]	2014	24507940	Exhcessive bleeding was defined as postoperative blood loss exceeding 1000 mL/12 h.
Orlov et al. [106]	2014	24445626	Patients whose calculated blood loss was part of the highest quartile for the cohort were classified as having had high blood loss. The amount of 1770 mL of CTD delineated high blood loss.
Doussau et al. [107]	2014	24117772	(1) Excessive intraoperative bleeding: Defined as either abnormal diffuse or microvascular bleeding uncontrolled by compression and electrocoagulation, needing blood transfusion of more than two units of RBCs or more than 400 mL of cell salvage blood during for patients weighing at least 60 kg (2) Excessive postoperative bleeding: Defined as a bleeding output of more than 1.5 mL/kg/h for at least 3 h or a need for surgical reexploration for hemostasis during the 48 postoperative hours.
Chowdhury et al. [108]	2014	24630471	Total CTD ≥ 600 mL within 12 h after surgery was used as a cut-off to define elevated CTD.
Dalen et al. [109]	2014	24447500	Excessive bleeding requiring reexploration was noted in patients with bleeding of more than 500 mL/h, or more than 300 mL/h in 2 consecutive hours coupled with hemodynamic instability
Ranucci et al. [110]	2014	25209096	Severe bleeding was defined as the presence of at least one of the following: CTD > 1 L in the first 12 postoperative hours, need for surgical reexploration, and need for >5 units of red blood cells of fresh frozen plasma.
Negargar et al. [111]	2014	25610554	Postoperative bleeding requiring intervention.
Agarwal et al. [112]	2014	25440634	Excessive bleeding requiring re-exploration occurred in the case of CTD greater than 500 mL in the first hour or greater than 1000 mL hin the first 4 h coupled with hemodynamic instability.
Totonchi et al. [113]	2014	25610551	Bleeding was defined as either transfusion of ≥5 U whole blood or packed red blood cells within a 48 h period or reoperation after closure of sternotomy for the purpose of controlling bleeding.
Fassl et al. [114]	2014	25324348	Major bleeding was defined as postoperative bleeding volumes >1000 mL during 24 h or the need for surgical re-exploration because of bleeding.
Espinosa et al. [115]	2014	25276093	Excessive bleeding was defined as persistent chest tube output >200 mL/h.
Kindo et al. [116]	2014	24857189	Excessive bleeding group was defined as patients with a 24 h chest tube output (CTD) exceeded the 90th percentile of distribution.
Kim et al. [117]	2014	24739221	Excessive bleeding requiring reoperation was noted when postoperative bleeding exceeded 200 mL/h for ≥6 h or ≥400 mL for the first 1 h.
Sharma et al. [118]	2014	24717423	Excessive bleeding was defined as chest tube output measured within 8 postoperative hours exceeded 75th percentile of distribution.
Ghavidel et al. [119]	2015	25587193	Abnormal or excessive mediastinal bleeding was defined as > 200 mL in a single hour or >1000 mL.
Mishra et al. [120]	2015	25566711	Excessive bleeding was defined as the amount of CTD exceeding 2.5 mL/kg/h within first 3 postoperative hours.
Drews et al. [121]	2015	24838516	Definition of excessive bleeding requiring reexploration was based on the discretion of the surgeon and intensivist and was based on triggers such as blood loss > 600 mL over the first hour, >400 mL for two consecutive hours, >300 mL for 3 consecutive hours coupled with unstable hemodynamic condition due to progressive chest blood loss.
Besser et al. [122]	2015	25440401	Excessive bleeding is defined as >2 mL/kg/h.
Lahtinen et al. [123]	2015	25281042	Excessive bleeding was defined as total postoperative drainage volume > 1000 mL.
Reed GW et al. [124]	2015	25655085	Excessive bleeding was defined in two ways: (1) The first way of defining excessive bleeding was based on the Thrombolysis in Myocardial Infarction (TIMI) definitions of bleeding, with major bleeding defined as clinically significant drop in Hgb of ≥5 g/dL or HCT ≥ 15%, or bleeding that resulted in death within 7 days. Minor bleeding was defined as hemorrhage resulting in a drop in Hgb of 3 ro <5 g/dL or HCT 9% to <15%. (2) The second way in defining excessive bleeding was based on the amount of postoperative chest tube output. Excessive chest tube output was defined as chest tube output within the top tertile, or >935 mL, within 24 h.

Abbreviations: CABG—coronary artery bypass graft; CPB—cardiopulmonary bypass; CTD—chest tube drainage; ICU—intensive care unit.

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
