# Peer review of "Methodological Considerations for Studies Evaluating Bleeding Prediction Using Hemostatic Point-of-Care Tests in Cardiac Surgery"

_jcm, 2024, doi:10.3390/jcm13226737_

Round 1
Reviewer 1 Report
Comments and Suggestions for Authors
I have read this manuscript with a great interest. Conducting a study on bleeding during cardiac surgeries is a challenging task. The complexity arises from the multifaceted nature of these factors, which include patient-related variables, surgical techniques, and perioperative management strategies. Each of these can significantly impact bleeding outcomes, making it difficult to isolate and study them individually. Therefore, research in this field requires meticulous study design.
-The subject is complex, and the article is written in a complicated language. The article frequently uses abbreviations, and the full forms of some of them are unclear.
-I am not sure if explaining the statistical analysis is necessary.
-I believe there is no need to explain study types (retrospective, prospective etc.) in detail.
-I appreciate the effort on the defining the weak and strong points of the current studies. The strengths and weaknesses of the studies in the literature have been briefly summarized. This effort should continue with defining specific suggestions about the methodology of further studies like the definition of bleeding and the tests (and times) etc.
-Making a list of suggestions instead of fig.1 should be considered.
-Conclusion section needs a revise. The statements in the conclusion are inconclusive. This section needs to be written more clearly.
Author Response
COMMENTS 1:
I have read this manuscript with a great interest. Conducting a study on bleeding during cardiac surgeries is a challenging task. The complexity arises from the multifaceted nature of these factors, which include patient-related variables, surgical techniques, and perioperative management strategies. Each of these can significantly impact bleeding outcomes, making it difficult to isolate and study them individually. Therefore, research in this field requires meticulous study design.
-The subject is complex, and the article is written in a complicated language. The article frequently uses abbreviations, and the full forms of some of them are unclear.
-I am not sure if explaining the statistical analysis is necessary.
-I believe there is no need to explain study types (retrospective, prospective etc.) in detail.
-I appreciate the effort on the defining the weak and strong points of the current studies. The strengths and weaknesses of the studies in the literature have been briefly summarized. This effort should continue with defining specific suggestions about the methodology of further studies like the definition of bleeding and the tests (and times) etc.
-Making a list of suggestions instead of fig.1 should be considered.
-Conclusion section needs a revise. The statements in the conclusion are inconclusive. This section needs to be written more clearly.
RESPONSE 1:
We greatly appreciate comments by Reviewer 1. Reviewer 1 pointed out some very important thoughts about studies evaluating bleeding during cardiac surgery. Multicausalty of bleeding presents challenge in understanding the risks for bleeding and identifying the causal relationship between bleeding risk factors and postoperative bleeding amount. We completely agree with reviewer 1, research in this field requires meticulous study design.
We double checked all abbreviations within the manuscript.
The authors organized soundingboard and we understand reviewer's concern whether statistical analysis is necessary. Please allow us to say how we feel it is needed to mention just basic statistical tests so to better understand the associations and causal relationship. Bleeding complications in cardiac surgery are persistent throughout decades and considering modern anticoagulant and antiplatelet drugs as well as evergrowing proportion of advance aged patients undergoing complex surgeries , we feel it is very important, for both clinicians and researchers to understand most relevant statistical tests to address the relevant questions.
Nowadays, we have in literature more and more retrospective analyses. In such analyses, the associations between ie. point-of-care tests and bleeding amount are tested. At the same time, point-of-care tests were regularly used as a part of clinical practice and that in fact presents a bias that hampers meaningful conclusions. On the other hand, retrospective analyses provide large sample studies which is important whenever analyzing infrequent outcome. It is really important to understand advantages and drawbacks of prospective and retrospective studies. Prospective studies may be controlled for potential bias, but usually have small study sample that makes them frequently underpowered. We believe it is of utmost interest to understand pros and cons for both retrospective and prospective studies. Needless to point out the importance of understanding the presence of intervention in studies (observational versus interventional study). When we put all aforementioned on research timeline we have a clear picture of research setting and we may better understand results of studies.
Conclusion section has been revised as per reviewer 1 suggestion.
Reviewer 2 Report
Comments and Suggestions for Authors
The authors present a comprehensive analysis of the various methodological pathways utilized for predicting bleeding in cardiac surgery. They examine different study designs, ranging from retrospective analyses to prospective cohort studies, and consider diverse patient populations, including high-risk groups and those with specific comorbidities. The review emphasizes the importance of standardizing key variables and outcome measures to ensure that studies are comparable and that their findings can be generalized across different clinical settings. By outlining best practices for study design, the authors aim to guide future research towards more robust and reliable predictions of bleeding risks, ultimately improving patient outcomes.
Minor revison: Please provide full words before using an abbrevation (for instance CBP, HCT...)
Comments on the Quality of English LanguageSatisfying
Author Response
Reviever 2 comments:
The authors present a comprehensive analysis of the various methodological pathways utilized for predicting bleeding in cardiac surgery. They examine different study designs, ranging from retrospective analyses to prospective cohort studies, and consider diverse patient populations, including high-risk groups and those with specific comorbidities. The review emphasizes the importance of standardizing key variables and outcome measures to ensure that studies are comparable and that their findings can be generalized across different clinical settings. By outlining best practices for study design, the authors aim to guide future research towards more robust and reliable predictions of bleeding risks, ultimately improving patient outcomes.
Minor revison: Please provide full words before using an abbrevation (for instance CBP, HCT...)
Authors response to reviewer 2:
We are grateful for Reviewer 2 comments that we found comments very valuable.
With regard to abbreviations we double checked their presence in the text and provided explanation where mentioned for the first time.
Reviewer 3 Report
Comments and Suggestions for Authors
Your submission represents a very extensive Review of information with analysis and citations from 150 sources.
The focus is mainly on comprehensive insight into the research field of bleeding complications in cardiac surgery. The authors have assumed that this Review will help a reader to better understand methodological flaws and how this influence current evidence.
However, after the reading a whole submission it becomes too difficult to identify the practical and more applicable site of this submission.
A better integrated and summarized information would serve much better and would be easier to use or would be more user-friendly to apply. The comparison between studies and reduction in the size of Tables (after integration of useful information) would make this submission more valuable.
Author Response
Reviewer 3 comments:
Your submission represents a very extensive Review of information with analysis and citations from 150 sources.
The focus is mainly on comprehensive insight into the research field of bleeding complications in cardiac surgery. The authors have assumed that this Review will help a reader to better understand methodological flaws and how this influence current evidence.
However, after the reading a whole submission it becomes too difficult to identify the practical and more applicable site of this submission.
A better integrated and summarized information would serve much better and would be easier to use or would be more user-friendly to apply. The comparison between studies and reduction in the size of Tables (after integration of useful information) would make this submission more valuable.
Authors response to Reviewer 3:
We greatly appreciate comment by Reviewer 3. We comprehensively reviewed literature and the manuscript in its present form is a sum of authors experience in this research field. Authors on byline have well over 22000 citations all together. Authors did their efforts to address this very important research field and to help both researchers and clinicians in conducting research and using the evidence in their clinical practice. It seems impossible to make this manuscript shorter and "easier to read" and to keep the necessary quality at the same time. We understand reviewer's comment, however the manuscript in its present form presents comprehensive review of literature rather than practice guideline.
Reviewer 4 Report
Comments and Suggestions for Authors
1. An interesting, comprehensive work discussing, on the basis of retrospective studies, the causes of postoperative bleeding in cardiology. Current guidelines and indications of the practical value of the research are also cited. If possible in retrospective studies, I suggest expanding the data on patients to include possible preoperative renal failure. Renal dysfunction can lead to platelet dysfunction, which increases the risk of bleeding.
Author Response
Reviewer 4 comments:
An interesting, comprehensive work discussing, on the basis of retrospective studies, the causes of postoperative bleeding in cardiology. Current guidelines and indications of the practical value of the research are also cited. If possible in retrospective studies, I suggest expanding the data on patients to include possible preoperative renal failure. Renal dysfunction can lead to platelet dysfunction, which increases the risk of bleeding.
Authors response to Reviewer 4 comments:
We greatly appreciate Reviewer 4 comments. Indeed, we agree with comment on the relationship between renal failure and platelets dysfunction. From our clinical practice we are aware that uremic platelets do have dysfunction to some degree. Literature evidence to this important research question already exists (ie. PMID 15497100 , PMID 24132242 ). In this present manuscript , we may say how the role of renal failure has been in fact quantified (therefore covered) throughout platelet function testing and/or thromboelastometry values. Indeed, we may go further in detail with this question, but that would require systematic review with meta-analysis and would go far beyond the scope of this paper addressing methodological considerations only. We greatly respect this comment and we plan to draft systematic review on the association between renal failure and haemostatis disturbances.
Round 2
Reviewer 1 Report
Comments and Suggestions for Authors
There are no further issues.
Author Response
Comments 1: There are no further issues.
Authors response: We would like to thank reviewer for the comment.
Reviewer 3 Report
Comments and Suggestions for Authors
This version of the submission is very similar to previous.
The authors did not reduce the size and did not follow the recommendation to condense the information and to make their submission more applicable.
Author Response
We greatly appreciate comment by Reviewer 3. We comprehensively reviewed literature and the manuscript in its present form is a sum of authors experience in this research field. Two authors on this manuscript co-authored the new 2024 Patient Blood Management guidelines published by EACTS/EACTAIC so we are aware of clinical and research importance of this manuscript and its reproducibility. We all agree that we cannot cut the table content and keep detailed information at the same time. We cannot change the manuscript text as per reviewer suggestion and keep our idea at the same time. That would require complete rework and would substantially change the idea of this manuscript. This is narrative review drafted by group of experts addressing very challenging clinical and research field. It is really not possible to shorten the manuscript text and keep all the relevant information and messages at the same time. In contrast to, some segments need further more detailed analysis, but that would require more text and as such would go far beyond the scope of the paper. Therefore, our research group has a plan to address some specific research questions throughout shorter concise systematic review with meta-analysis, but this is completely new manuscript and different type of manuscript. For sure, this research field has a lot of questions to address. Our research group believes that the best way for this present manuscript is to keep it in present form so to keep the original idea, considerations and messages. Authors on byline have well over 22000 citations all together and we believe this manuscript in its present form would present valuable contribution.
Authors did their efforts to address this very important research field and to help both researchers and clinicians in conducting research and using the evidence in their clinical practice.
It seems impossible to make this manuscript shorter and "easier to read" and to keep the necessary quality at the same time.
We understand reviewer's comment, however the manuscript in its present form presents comprehensive review of literature rather than practice guideline.